# Estimation of Position and Size of a Contaminant in Aluminum Casting Using a Thin-Film Magnetic Sensor

**DOI:** 10.3390/mi13010127

**Published:** 2022-01-14

**Authors:** Tomoo Nakai

**Affiliations:** Industrial Technology Institute, Miyagi Prefectural Government, Sendai 981-3206, Japan; nakai-to693@pref.miyagi.lg.jp; Tel.: +81-22-377-8700

**Keywords:** magnetic thin film, magneto-impedance sensor, high frequency, surface-normal magnetic field, nondestructive inspection, tool steel chipping

## Abstract

Advanced manufacturing processes require an in-line full inspection system. A nondestructive inspection system able to detect a contaminant such as tool chipping was utilized for the purpose of detecting a defective product as well as damaged machine tools used to fabricate the product. In a previous study, a system able to detect magnetic tool steel chipping in conductive material such as aluminum was developed and tested. In this study, a method of position and size estimation for magnetic chipping was investigated and is described. An experimental confirmation of the proposed method was also carried out using an actual prototype system.

## 1. Introduction

Inspection of all items in the manufacturing process is desirable for advanced manufacturing systems, aiming at a reduction of products’ defects, the detection of damaged machine tools, and the tuning of the processing conditions. Nondestructive detection of magnetic contaminants such as a tool steel chipping has been studied using thin-film magnetic sensors [1,2]. Thin-film magneto-impedance (MI) sensors [3,4,5,6,7,8] have high sensitivity and are used for detecting magnetic small chipping nondestructively [9,10,11]. A thin-film magneto-impedance (MI) sensor made of a single-layer amorphous film was designed and developed previously [12]. It aims to realize a nondestructive detection of magnetic small contaminants in aluminum casting. In this system, an array of the thin-film sensors was set in a strong and uniform magnetic field, with the sensing direction perpendicular to the magnetic field direction. This system can detect a magnetic field leaking from the contaminant with high sensitivity, simultaneously with the magnetization of the contaminant by a strong static magnetic field. If the static magnetic field present a non-uniformity in the sensing area, an eddy current is generated in the aluminum casting while feeding the measured work piece in it. A structure with a uniform magnetic field and a high-sensitivity driving circuit of the sensor was realized, suitable for controlling the setting position and direction in the strong field [1]. A prototype detection system was constructed, and its performance in detecting magnetic contaminants was confirmed [12].

Position detection using a magnetic field was reported using a combination of excitation coil and sensing marker coils or a magnet and 3-dimensional magnetic sensors. The excitation coil or magnet generates a magnetic field in the measurement area, and the sensing apparatus detects the 3-dimensional field and estimates the position of the marker or magnet [13,14]. They use a convergence calculation using an iteration method. Position detection suitable for a manufacturing system using a feeding conveyer has not been studied yet. In this study, a method of estimating position and size of a magnetic contaminant, suitable for the structure of our previously reported system, is proposed and experimentally verified.

## 2. Concept and Method of Estimation

In this section, the concept at the basis of the developed measurement system is explained.

Figure 1 shows a schematic explanation of the concept at the basis of the measurement system. The system was set on a feeding conveyer of workpiece and could nondestructively detect the presence of contaminants and estimate their position and size. This proposed system can realize a full inspection of a non-magnetic metal workpiece for the detection of magnetic contaminant chipping.

Figure 2 shows a picture of the prototype measurement system, which has a magnetic structure applying a certain vertical magnetic field to the measurement area and a thin-film magneto-impedance sensor array that, with magnetization, simultaneously detects magnetic chipping in a bulk workpiece. The system is installed with belt conveyer for feeding the measurement workpiece.

In this study, a method of estimating the chipping position and size inside the workpiece is proposed, which is suitable for the layout of the measurement system shown in Figure 2. Figure 3, Figure 4 and Figure 5 report schematic explanations of the structure of position estimation system proposed in this article. Figure 3 shows a perspective view. Figure 4 shows a side view, and Figure 5 shows a top view. In these figures, a homogenized vertical magnetic field was established between two soft magnetic plates having a high saturation magnetization. High-sensitivity thin-film MI sensors were placed on the surface of the homogenization plate, and the workpiece, including a magnetic small chipping, was fed transversally along the X-direction. In this system, the MI sensors were single-axis, and the sensing direction was placed parallel to the X-direction. The conveyer fed the workpiece along the X-direction at a constant feeding speed. In this system with limited dimensions, the 3-dimensional position of magnetic chipping running through the feeding line could be estimated analytically, which was the aim of this study.

The fundamental equation governing this operation is the Equation for a magnetic dipole:(1)$$H=-\frac{1}{4\pi \mu_{0}}\nabla \left(\frac{m\cdot r}{r^{3}}\right)$$
where ***m*** is the magnetization of the dipole (*m*_x_, *m*_y_, *m*_z_). *r* = (*x*, *y*, *z*) indictaes a position in which the magnetic field ***H*** coming from the magnetic dipole ***m*** is estimated. In this study, magnetic chipping having a certain volume was assumed to be a magnetic dipole with the same magnetization ***m***.

When dimensional limitations were assumed as shown in Figure 3, Figure 4 and Figure 5, the dipole Equation was transformed in Equation (2), in air:(2)Bx=3mxz4π(x2+y2+z2)52

In this transformation, both ***m*** = (0, 0, *m*_z_) and the constants (*y*, *z*) were assumed to be subjected to the directional limitation of the measurement system, in which soft magnetic chipping was fed parallel to the X-axis within the magnetic field, which had a vertical direction. The reason why the component of the estimated magnetic field was only in present on the X-axis was linked to the sensing direction of the sensor installed in the vertical magnetic field, as shown in Figure 4.

Based on the above Equation, sensing waveforms were numerically estimated as shown in Figure 6. This indicated feeding dipole waveforms that were detected by three sensors placed in different positions. Each waveform had one minimum point and one maximum point, and the zero-crossing point between them indicated that the dipole was placed just above the measuring sensor element.

The aim of this estimation of the magnetic dipole’s position was the detection of the position of extreme points appearing in the measured waveform. Extreme positions were identified by the Equation d*B*_x_/d*x* = 0, using Equation (2).

Figure 7 shows an example of variation of the maximum extreme point when a magnetic dipole was fed along the dashed line on *y* = 50. The height of this feeding line of the dipole was *z* = 50. This figure shows a dependence of the X-position of the extreme point on the Y-position. When a sensor was placed on the position (*x*, *y*, *z*) = (0, 0, 0), the extreme point of the measured magnetic flux density *B*_x_ was observed at the dipole position *x* on the feeding line, which was at the same *x* value of the crossing point of the dotted line *y* = 0 with the solid curve. When a sensor was placed on the position (0, −50, 0), the extreme point of *B*_x_ was observed at the dipole position *x* on the feeding line, which was at the same the *x* value of the crossing point of the dotted line on *y* = −50 with the solid curve. When a sensor was placed on the position (0, 50, 0), which was just below the feeding line of the dipole, the extreme point was observed at the dipole position *x* on the feeding line, which was at the same *x* value of the crossing point of the dotted line on *y* = 50 with the solid curve. In the last case, the extreme point was observed at *x* = *H*/2, where *H* is the height of the feeding line, in this case, *x* = 25.

Figure 8 is the analogous estimation of the extreme point when the feeding line passed the point (*y*, *z*) = (50, 100), which was 50 mm higher than the feeding line, as shown in Figure 7. In this case, the extreme point appeared at higher value of *x* compared with its position in Figure 7. From these figures, it is expected to estimate the feeding position (*y*, *z*) based on the dimensions of extreme points obtained by several sensors.

Based on Equation (2), the first-order differentiation of *x*, when the (*y*, *z*) was constant, was easily derived as follows:(3)dBxdx|(y, z)=const.=ddx{3mz·z4π·x(x2+y2+z2)−52}      =3mz·z4π(x2+y2+z2)−72×{(y2+z2)−4x2}
when *B**_x_* had the extreme value, the first derivative equaled zero:$$\frac{dB_{x}}{dx}=0$$

Based on Equation (3), the following equation was derived at the extreme point:(4)$$4x^{2}=y^{2}+z^{2}$$

From the Equation (4), it is understood that the extreme point *x* is derived from the sectional position (*y*, *z*), independent of the magnetization *m*_z_.

By using two Equations of extreme points which were obtained by two sensors, one placed at (0, 0, 0) and the other placed at (0, *DS*, 0), Equation (4) results in two Equations, as follow:(5)$$4\cdot XS1^{2}=y^{2}+z^{2}$$
(6)$$4\cdot XS2^{2}={\left(y-DS\right)}^{2}+z^{2}$$
where sensor-1 (indicated by *S*1) was placed at (0, 0, 0), and sensor-2 was placed at (0, *DS*, 0). The extreme point which was measured by *S*1 is indicated as *XS*1, and the point measured by *S*2 is indicated as *XS*2.

By erasing *z*^2^ using Equations (5) and (6), the feeding position (*y*, *z*) is derived analytically as follows:(7)$$y=\frac{1}{2}\times \left\{\frac{4\left(XS1+XS2\right)\left(XS1-XS2\right)}{DS}+DS\right\}$$
(8)$$z=\sqrt{4\times XS1^{2}-y^{2}}$$

Based on the sensing waveform in Figure 6, the zero-crossing point appeared when the magnetic dipole was placed just above the sensor. The extreme points appeared symmetrically with respect to this zero-crossing point, then *XS*1 and *XS*2 were independent of the maximum or the minimum. It is possible to estimate this value as follows:(9)$$XS\#=\frac{XS{\#}_{MAX}-XS{\#}_{min}}{2}$$
where # is 1 or 2, indicating the sensors, the suffix “MAX” indicates the maximum point, and the suffix “min” indicates the minimum point. Equation (9) is useful when the sensor is not placed at the original point or when each sensor has a different X-position.

Figure 9 shows a flow chart of position estimation based on our proposed equations. Some experimental confirmations were carried out and are presented in the following sections, based on this flow chart.

## 3. Experimental Confirmation

Experimental confirmation of the proposed method of position estimation was carried out and is reported in this section.

Figure 10 shows a schematic of the experimental apparatus of this study. This measurement apparatus was described in another article previously published [12]. In this apparatus, magnets are fixed on the inner surface of a rectangular magnetic core to generate a strong vertical field in the measurement area. Soft magnetic plates are put on both the upper and the lower magnets to make the field uniform. They are called homogenization plates. Sensors are set on the surface of the lower homogenization plate, near the middle position of the plate on the feeding direction X.

Figure 11 shows a side view of magnetic field distribution, a contour line expression of the vertical magnetic flux density *B*_z_. The homogenization plates can produce a uniform field, but a certain degree of expansion would appear around the middle area of the measurement area. The vertical magnetic flux density was 37 mT at the sensor position and 24 mT in the middle of the measurement area. This apparatus can detect a soft magnetic small piece when it is fed along the X-direction in the measurement area between the homogenization plates.

The sensor element was fabricated by a thin-film process. An amorphous Co_85_Nb_12_Zr_3_ film was RF-sputter-deposited onto a soda glass substrate and then micro-fabricated into rectangular elements by a lift-off process. The element was 1000 μm in length, 50 μm in width, 2.1 μm in thickness. Tens of elements were aligned in a parallel configuration and connected by Cu thin-film strips to form a meander pattern. A magnetic field was applied during RF-sputter deposition for the purpose of inducing uniaxial magnetic anisotropy. Magnetic anisotropy in this study was oriented in the width direction, that is, the short-side direction of the element strip. It was induced by the direction of the magnetic field while sputtering. The element and the fabrication process were the same as in our previous work [1].

Figure 12 shows the variation of the sensor output as a function of the applied magnetic flux density along the sensing direction. The sensor was driven by a 400 MHz current and detected using a logarithmic amplifier [1]. In this sensor, the output voltage of 100 mV corresponded to 0.1 μT (1 mG). The following measurements were obtained by using sensors which had the same configuration as the presented sensor system.

Figure 13 shows one of the measurement samples made of tool steel. Its dimensions were *ϕ*5 × H4 mm. The two experimental confirmations described in this article used this sample.

### 3.1. Height Estimation Just above the Sensor

Figure 14 is a schematic explanation of the measurement layout for the first experiment. Figure 14a shows a front view and Figure 14b shows a side view of the layout. The front view is shown from the viewpoint of the feeding sample. The sample was fed at constant velocity, 100 mm/s, along the X-axis just above the middle sensor of the installed three sensors. In this case, the sample height *H* was the parameter to be estimated.

Figure 15 is a numerically simulated variation of *B*_x_ on the sensor position as a function of X. The magnetic dipole passed just above the sensor at the origin. Two extreme points appeared symmetrically with respect to the zero-crossing point, one was the maximum point, and the other was the minimum point. The extreme position *XS*1 was derived using Equation (9) and two extreme points. In the case in which the dipole was running through just above the sensor, the height was estimated as *H* = 2 × *XS*1 based on Equation (8). This Equation also indicated that the feeding height *H* equaled the distance between two extreme points, which was between the maximum and the minimum points.

Figure 16 and Figure 17 show examples of the measured data. Figure 16 shows the case in which the feeding height was *z* = 32 mm, and Figure 17 that in which the feeding height was *z* = 47 mm. In these figures, the horizontal axis indicates time, and the sample was fed in the +X-to-−X direction, which is the reason why these figures have a opposite polarity compared with Figure 15. In comparison with Figure 16 and Figure 17, the horizontal distance of the two extreme points in Figure 17 had a larger value. This is because the horizontal distance between the two extreme points, i.e., the maximum point and the minimum point, corresponded to the feeding height of the dipole from the sensor element, which was deduced from Equation (8). The output signal 1 V corresponded to 10^−6^ T (10 mG) based on Figure 12, and the time of 0.1 s on the horizontal axis corresponded to 10 mm based on the feeding velocity. These are raw data; therefore the zero-crossing point was not set to be at an original point.

Figure 18 shows the dependence of the maximum value of sensor output voltage on the height of tool steel chipping. From Equation (2), the measured magnetic flux density decreased as the inverse of the cubic function of distance. This figure shows a tendency, as expected according to this Equation. The measured sample was *ϕ*5 × H4 mm (Figure 13).

Figure 19 shows an experimental result of height estimation. The horizontal axis represents the actual height of feeding tool steel chipping, and the vertical axis represents an estimated height based on Equation (8). This estimation of height, just above the sensor, had a measurement error that increased as the height increased. The reason of this increment of the measurement error is that the magnetic flux contour line around the middle height of the measurement area expanded toward the outer direction. This resulted in a variation of the direction of the magnetic flux vector from that of parallel vertical lines; therefore, it induced an uneven magnetic field in the measurement area. Based on the measured results of Figure 11, the contour line of magnetic flux expanded outward around the middle of the height; then, the magnitude of the flux density also decreased as a function of the distance from the sensor. This effect was expected to change the position of the measured extreme point towards an inner position. This non-uniformity of the strong applied magnetic field could induce an estimation error not only for height position Z but also for the width position Y.

### 3.2. Position Estimation in Constant Height z = 54 mm

Figure 20 shows the layout for the second experiment. In this case, the width position Y was at a constant height *z* = 54 mm. Figure 20a shows a front view, and Figure 20b shows a side view of the feeding line in this experiment. The front view is shown from the viewpoint of the feeding sample. The feeding velocity was 70 mm/s in this experiment.

Figure 21 shows the measured waveforms of three sensors when the feeding line was at *y* = 10 and *z* = 54. Figure 22 shows the waveforms when *y* = 30, *z* = 54. Compared with these waveforms, the magnitude of the extreme points in the largest and the second larger line differed greatly, as shown in Figure 18. In this measurement, a certain degree of noise appeared, due to the detection electric circuit. The detection of the extreme point in the second larger line suffered a measurement error due to this electric noise. In this measurement, the polarity of the sensor driving circuit was set as negative.

Table 1 shows the results of position estimation. Δ*r* in this table is the estimated error. Figure 23 is a schematic illustration of the result. In the figure, the actual positions of the feeding line of the magnetic piece are shown as red points in the Y–Z plane, and their estimated feeding positions are shown as blue points. The correspondence is indicated by arrows in the figure. This result shows that the points which were adjacent to the right overhead of a sensor could not be precisely estimated, reporting an 8 mm to 10 mm estimation error. This is because the sensor signal with the second larger extreme point decreased in magnitude, almost to the same level as the noise of the detection circuit. The results showed that the feeding point placed almost in the middle of the two sensors could be estimated with good precision, and its estimation error was smaller than the sample size of *ϕ*5 × H4 mm. Estimation accuracy may be enhanced by increasing the sensitivity of the sensor and decreasing the circuit noise. Increasing the applied magnetic field can also be effective.

### 3.3. Position and Size Estimation in an Arbitrary (y,z) Position

Figure 24 shows three different specimens used for the final experiment. In this experiment, tool steel chippings with different sizes were placed in different feeding positions. In this experiment, both the feeding position (*y*, *z*) and the magnetic moment *m* were estimated experimentally. The feeding velocity was 70 mm/s. In Figure 24, the dimensions and weight of the samples are shown; they are as follows:

Sample 1: *ϕ*5 × H4, 0.63 g

Sample 2: *ϕ*6 × H4 with *ϕ*2 central hole, 1.11 g

Sample 3: *ϕ*5 × H10, 1.59 g

Table 2 shows the results, schematically reported in Figure 25. The actual positions of the feeding line of the magnetic piece are shown as red points in the Y–Z plane, and their estimated feeding position are shown as blue points. The correspondence is indicated by arrows in the figure, as for the previous experiment. In this case, the position was estimated with good precision, because all samples were not in the area adjacent to the right overhead of the sensor. The estimation error was smaller than the sample size. The magnitude of the magnetic moment *m* approximately corresponded to the volume of the sample. It is well known that the magnetic moment is affected by a demagnetizing force when a soft magnetic sample is magnetized in a constant magnetic field. In this study, even when the effect was taken into consideration, the order of magnitude of the magnetic moment was in accordance with the sample volume. The estimated order of the magnetic moment is shown in Table 2. It is shown that the estimated order of the magnetic moment was in agreement with the order of the sample volume.

## 4. Discussion

Our proposed method to estimate the position of a magnetic contaminant was applied to actual measurements. We showed that it is independent of the sensitivity and polarity of each sensor, because it is based on position detection of the extreme point.

In this section, a method of curve fitting is described for the purposes of improving the accuracy and carrying out an automatic detection of the extreme position. The fundamental Equation in this case is the previously presented Equation (2). Based on this Equation, a method of extracting feature quantity is proposed, and curve fitting against the measured sensor waveform is tried and discussed.

Based on the introduction of certain feature quantities, Equation (2) is transformed as follows:(10)Bx=A×(x−B){(x−B)2+P′}52+C
where *A*, *B*, *C*, *P*′ are parameters for curve fitting. *A* is a coefficient of magnitude of the waveform. *B* represents the X-positional displacement of the measurement sensor. *C* is a measurement offset of the magnetic flux density *B*_x_. *P*′ is as follows:(11)$$P^{\prime }=y^{2}+z^{2}$$

Based on Equation (4), the extreme position *x*_p_ is deduced as follows:(12)$$x_{p}=\sqrt{\frac{P^{\prime }}{4}}$$

It is easily understood that the extreme position is obtained from the measured waveform using the fitting parameter *P*′.

Figure 26 shows an example of curve fitting for a measured result. The dotted line is the measured result of the feeding position (*y*, *z*) = (10, 54). The fitting was carried out using a function contained in Origin 2021. Even if the fitting procedure was completed, a certain mismatch was observed in this result. The estimated extreme position obtained by curve fitting was different from the measured waveform. A possible reason is that the magnetic structure of this experiment had a certain distortion compared with an ideal parallel normal field.

Figure 27 indicates the difference of the obtained peak point between the raw data and the curve-fitting data. The difference was almost 10%, and curve-fitting underestimated it in this case.

## 5. Conclusions

A method for the nondestructive estimation of position and size of a magnetic contaminant in conductive materials such as aluminum casting was proposed. Based on the equation of a magnetic field generated by a magnetic dipole, a simple equation for estimating position and size of magnetic contaminant was derived. The limitation of dimensions and directions of the measurement system, in which the magnetization direction lies in the Z direction, the feeding direction of the contaminant is the X direction, and the magnetic sensor detects a magnetic field in the X direction, makes the fundamental equation to be a simple analytic equation. Experimental confirmations were also carried out in this study, and a certain accuracy was obtained. The estimation accuracy changed depending on the sample position relative to the sensor position. The estimation error was smaller than the sample size when the sample was fed through the area in the middle of two sensors, whereas it was twice as larger when it was fed through the area just above the sensor. In order to improve the accuracy of the estimation, a low-noise driving circuit and a highly uniform and wide normal magnetic field are needed in the measurement system. Magnetic field distribution also caused an eddy current inside the measured workpiece and caused a measurement noise for this detection system. We have already established another magnetic apparatus with a wide and uniform magnetic field and will described its enhancement in a future study.

## Figures and Tables

**Figure 1 micromachines-13-00127-f001:**
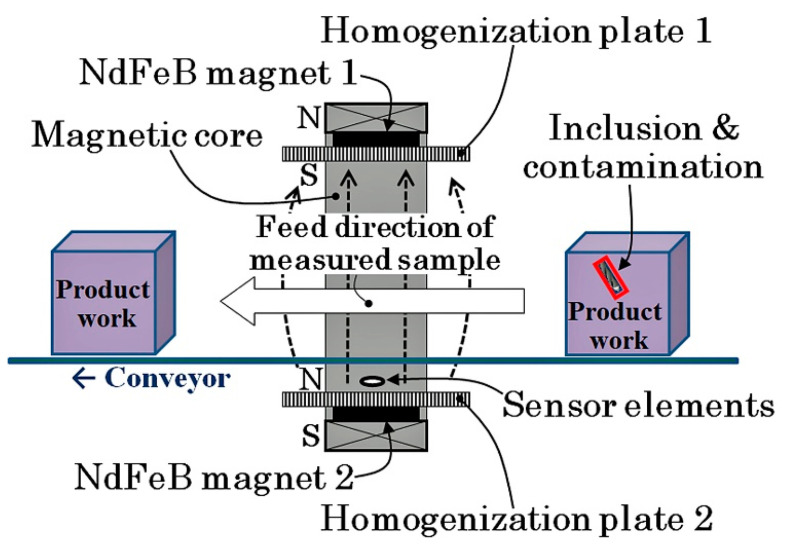
Schematic illustration of the concept at the basis of the measurement system (from [12]).

**Figure 2 micromachines-13-00127-f002:**
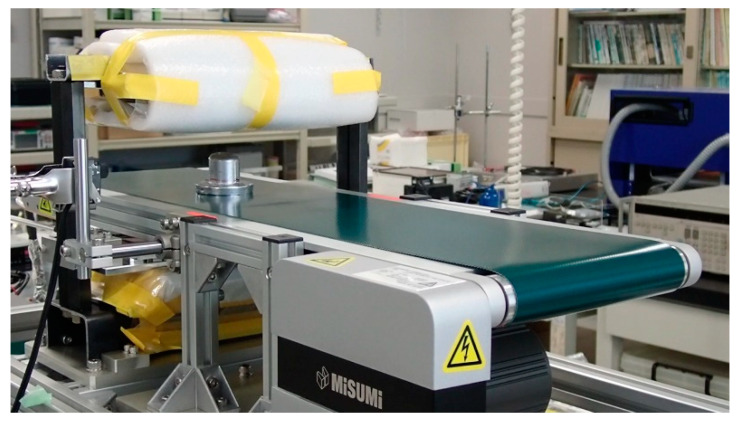
Picture of the prototype system with the conveyer.

**Figure 3 micromachines-13-00127-f003:**
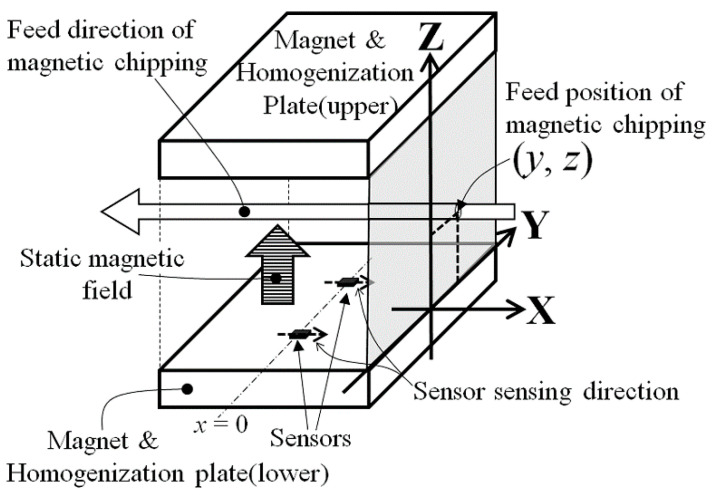
Perspective view of the structure of the position estimation system.

**Figure 4 micromachines-13-00127-f004:**
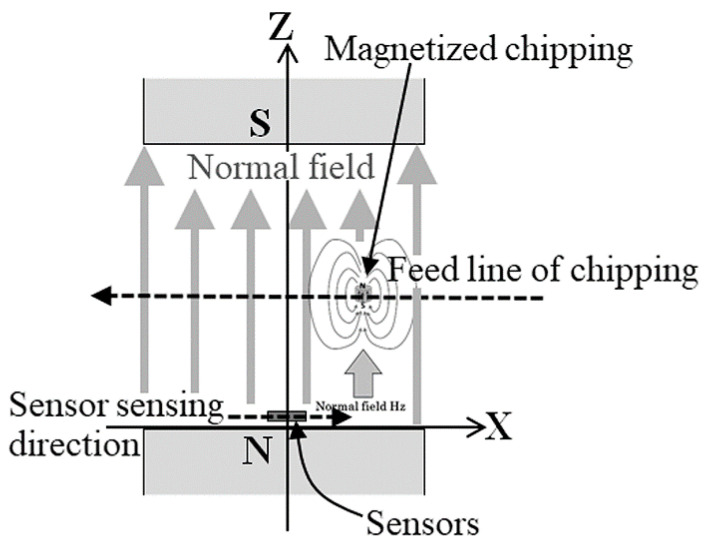
Side view of the structure of the position estimation system.

**Figure 5 micromachines-13-00127-f005:**
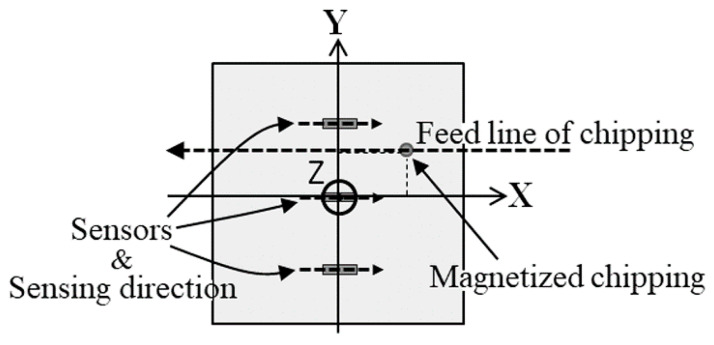
Top view of the structure of the position estimation system.

**Figure 6 micromachines-13-00127-f006:**
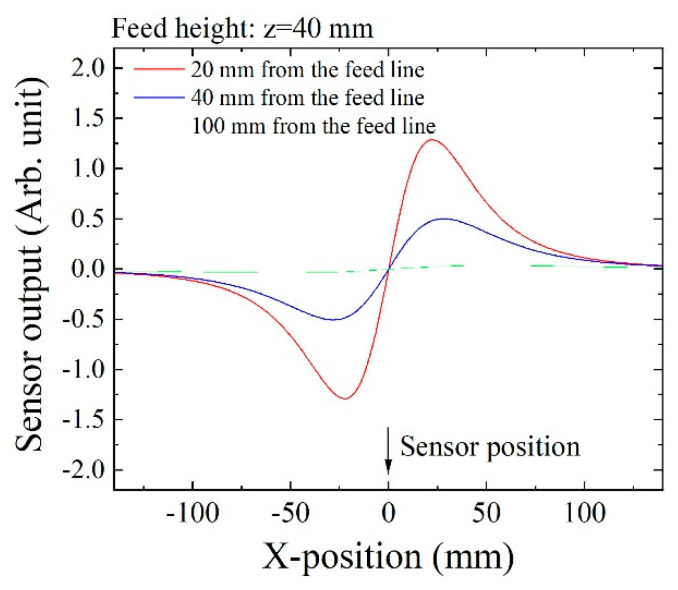
Numerically estimated sensing waveforms.

**Figure 7 micromachines-13-00127-f007:**
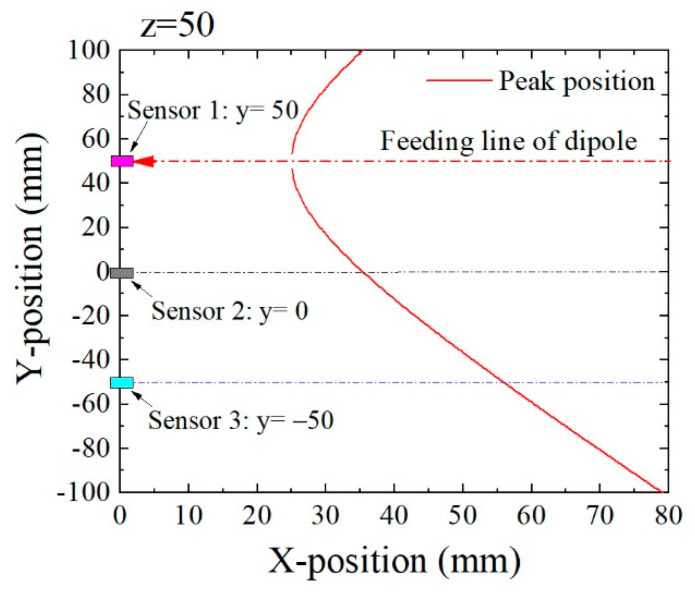
Variation of the extreme point’s position when a magnetic dipole was fed on *y* = 50, *z* = 50.

**Figure 8 micromachines-13-00127-f008:**
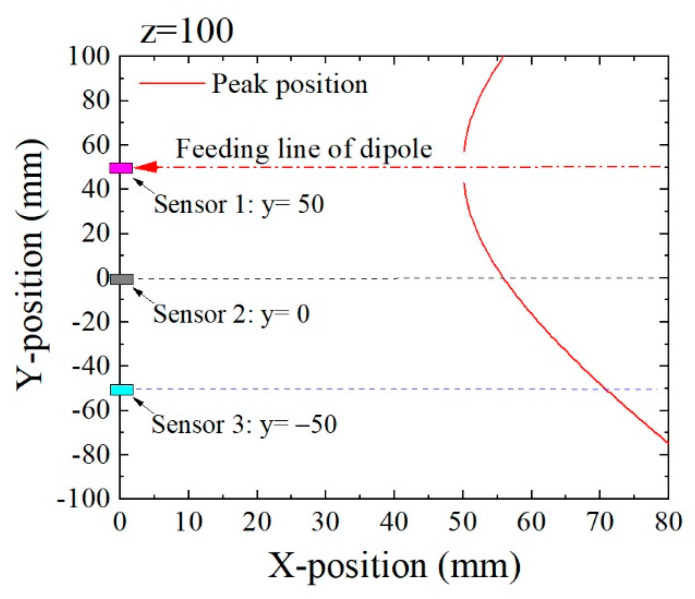
Variation of the extreme point when a magnetic dipole was fed on *y* = 50, *z* = 100.

**Figure 9 micromachines-13-00127-f009:**
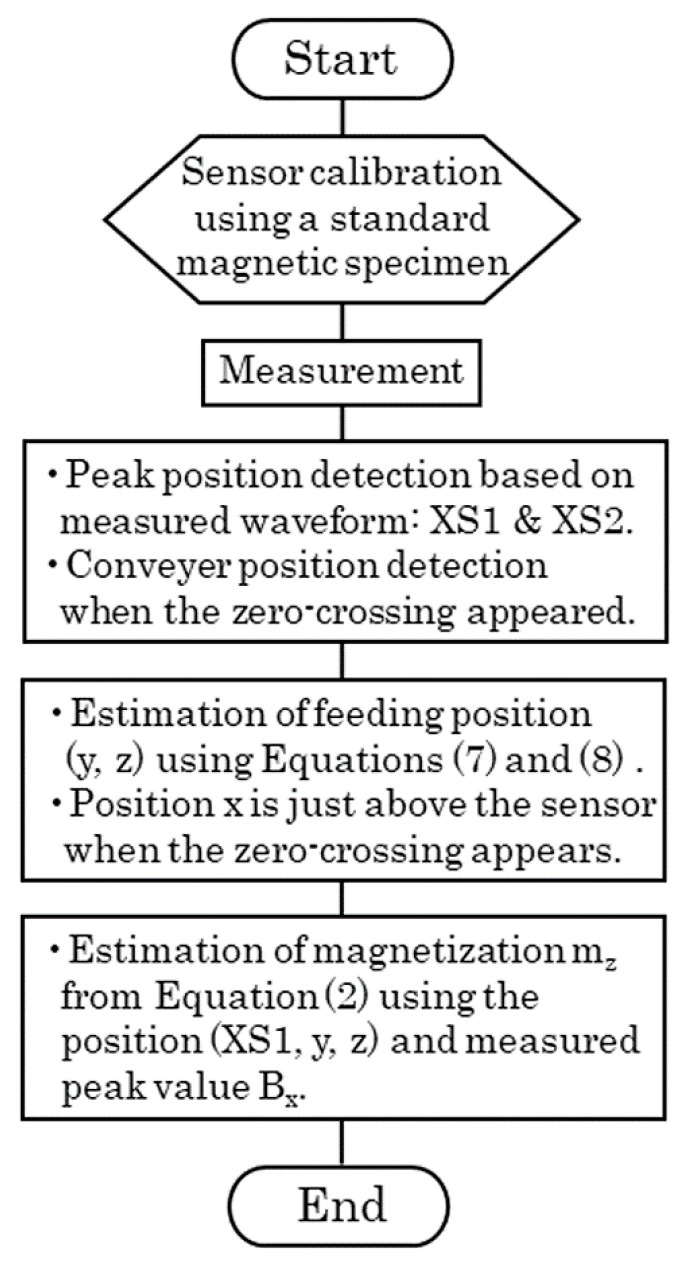
Flow chart of the proposed position estimation method.

**Figure 10 micromachines-13-00127-f010:**
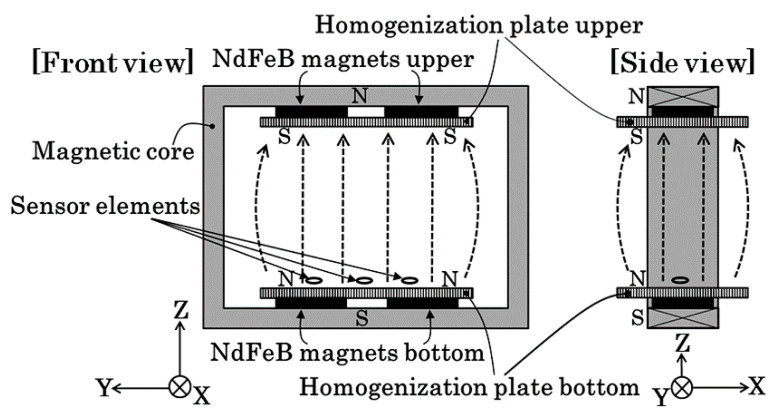
Schematic of the experimental apparatus (from [12]. The coordinates are revised to fit this paper).

**Figure 11 micromachines-13-00127-f011:**
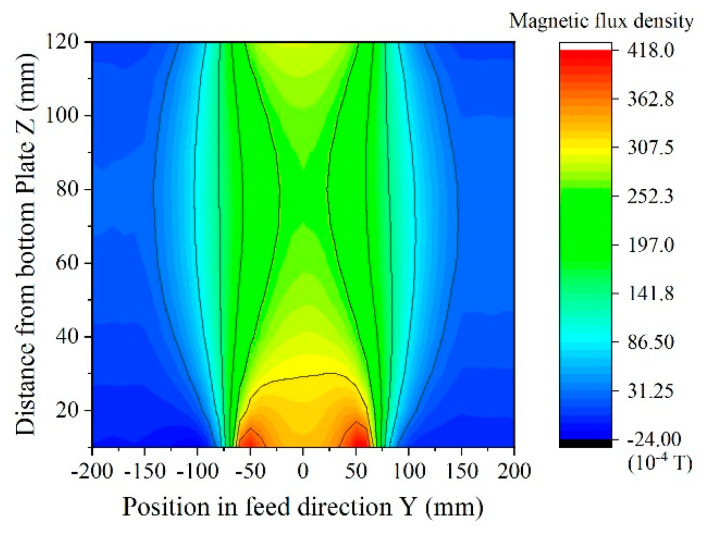
Side view of magnetic field distribution (from [12]).

**Figure 12 micromachines-13-00127-f012:**
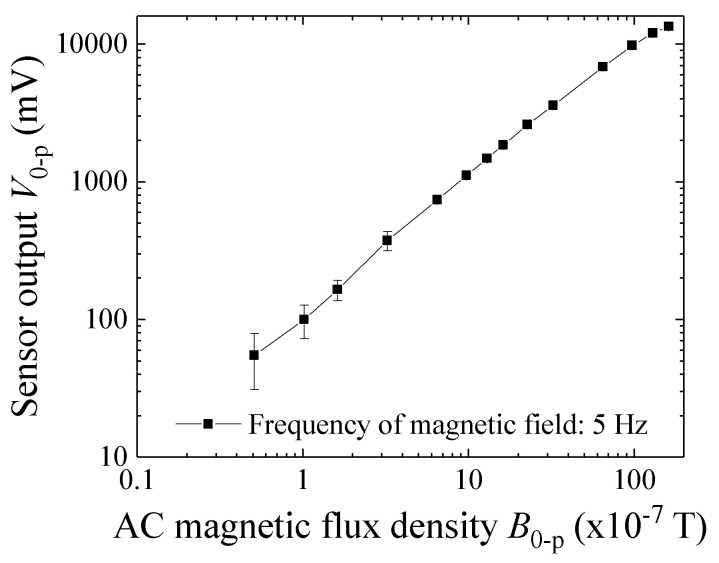
Variation of the sensor output as a function of magnetic flux density (from [1]).

**Figure 13 micromachines-13-00127-f013:**
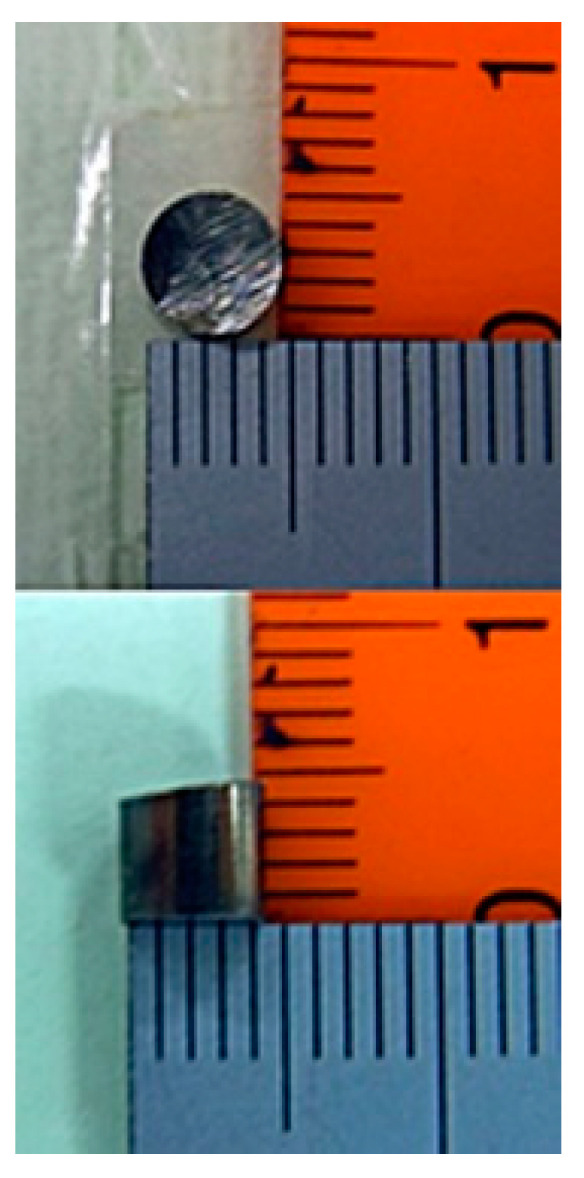
One of the measurement samples made of tool steel.

**Figure 14 micromachines-13-00127-f014:**
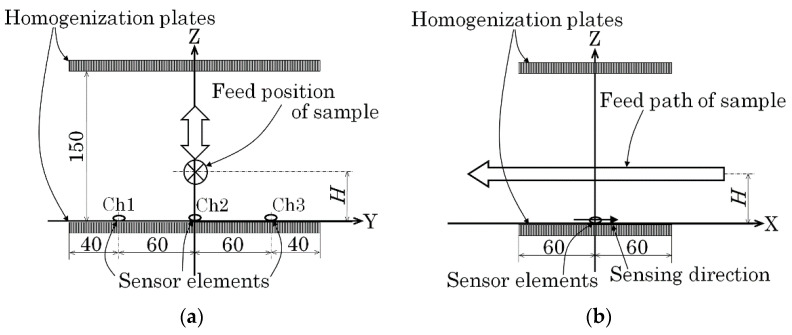
Schematic explanation of the measurement layout (first experiment). (**a**) Front view (View point of feeding sample). (**b**) Side view.

**Figure 15 micromachines-13-00127-f015:**
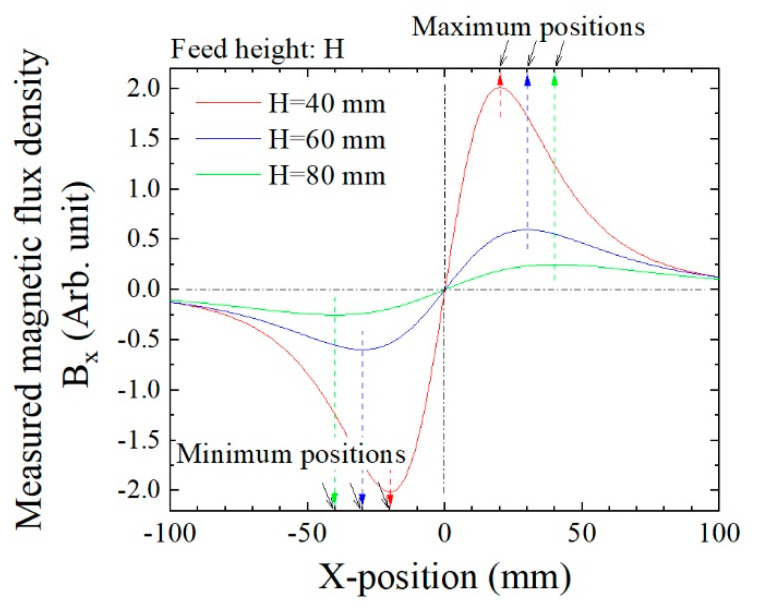
Schematic explanation of extreme points obtained by numerical simulation.

**Figure 16 micromachines-13-00127-f016:**
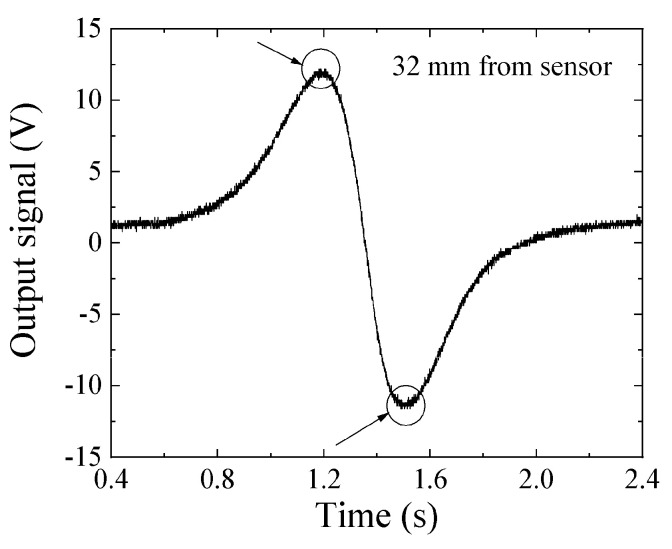
Measured waveform when *z* = 32 mm.

**Figure 17 micromachines-13-00127-f017:**
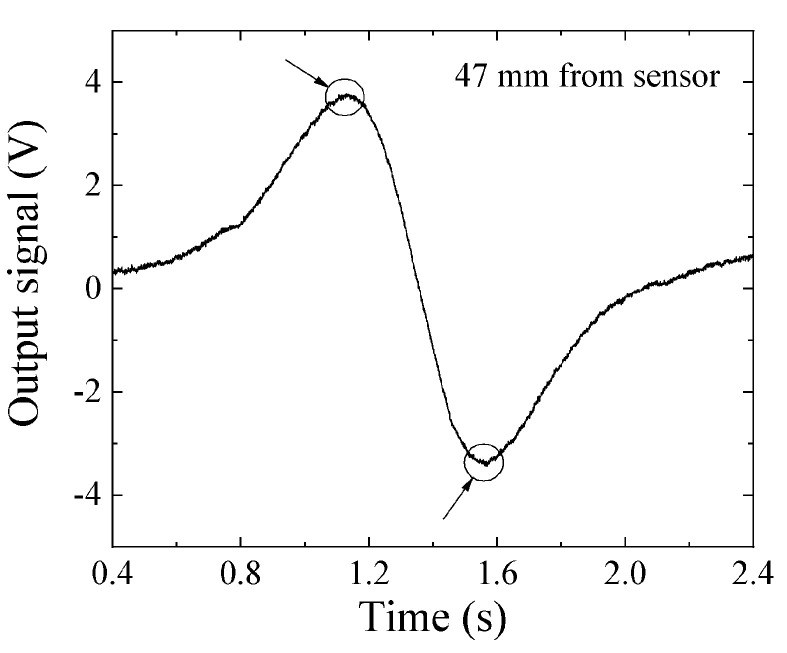
Measured waveform when *z* = 47 mm.

**Figure 18 micromachines-13-00127-f018:**
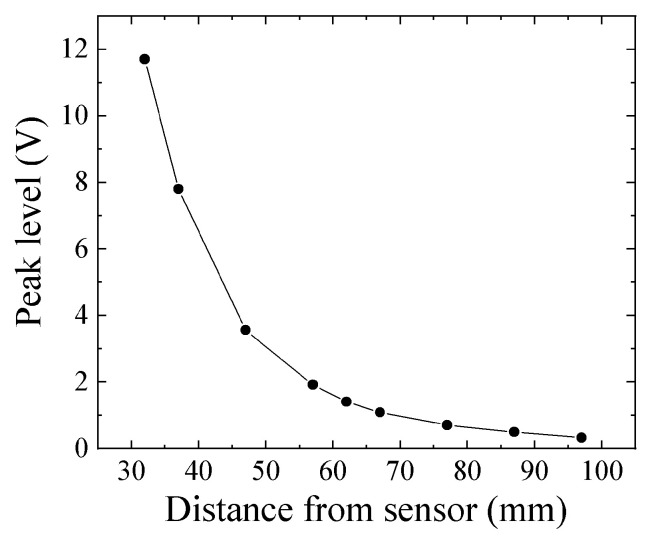
Dependence of sensor output voltage on tool steel chipping height.

**Figure 19 micromachines-13-00127-f019:**
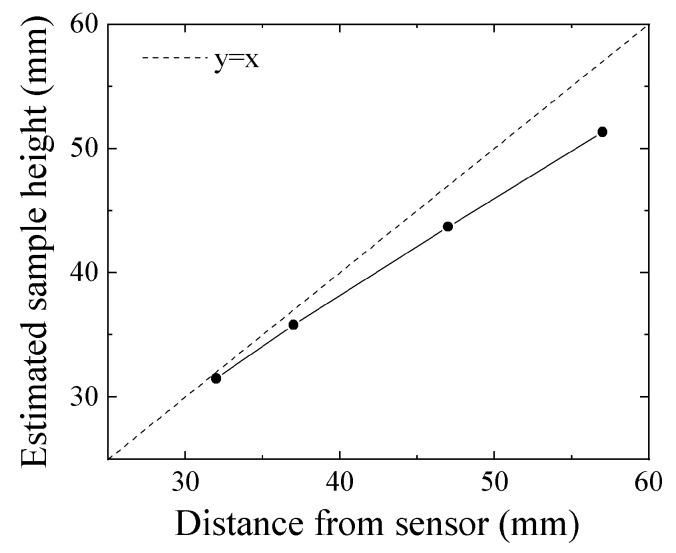
Experimental results of height estimation.

**Figure 20 micromachines-13-00127-f020:**
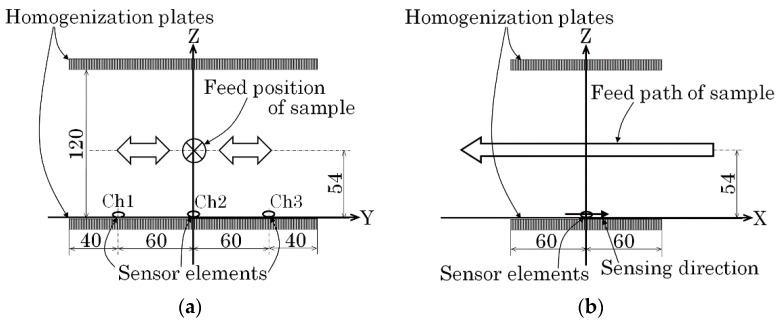
Schematic representation of the measurement layout (second experiment). (**a**) Front view (View point of feeding sample). (**b**) Side view.

**Figure 21 micromachines-13-00127-f021:**
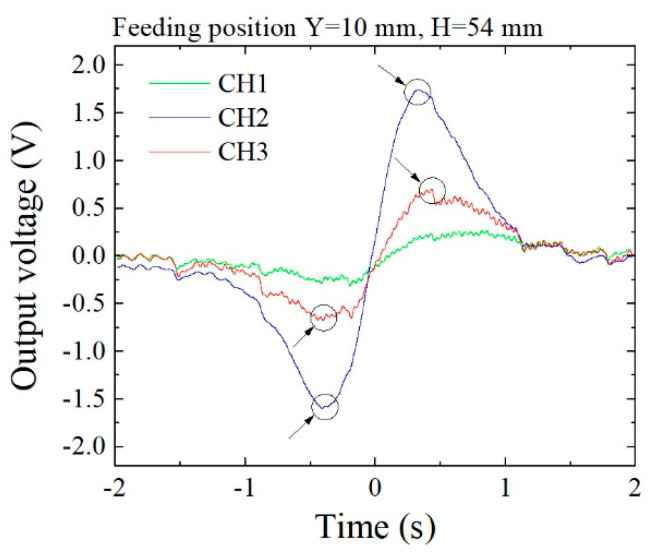
Measured waveforms (feeding point: *y* = 10, *z* = 54).

**Figure 22 micromachines-13-00127-f022:**
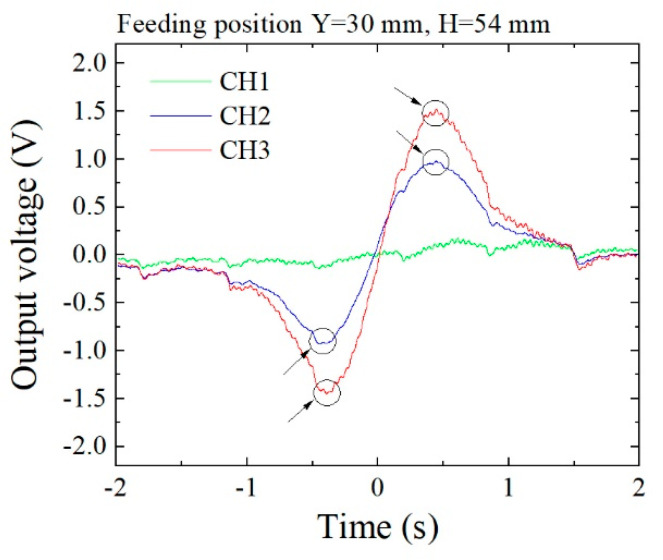
Measured waveforms (feeding point: *y* = 30, *z* = 54).

**Figure 23 micromachines-13-00127-f023:**
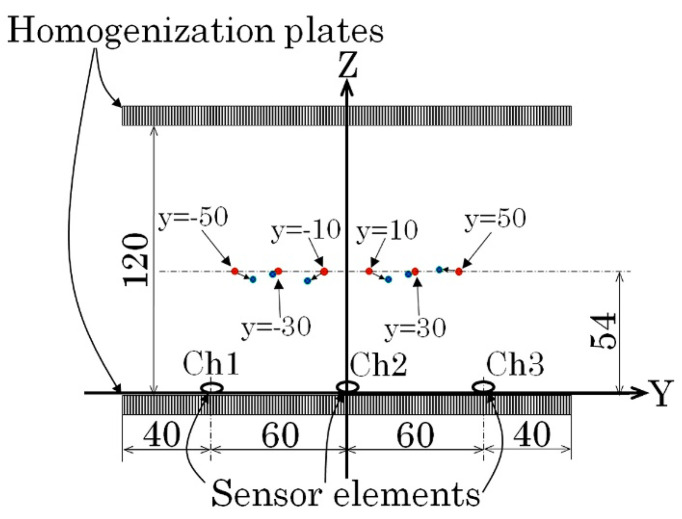
Estimated results for the second experiment (Red: actual position, Blue: estimation).

**Figure 24 micromachines-13-00127-f024:**
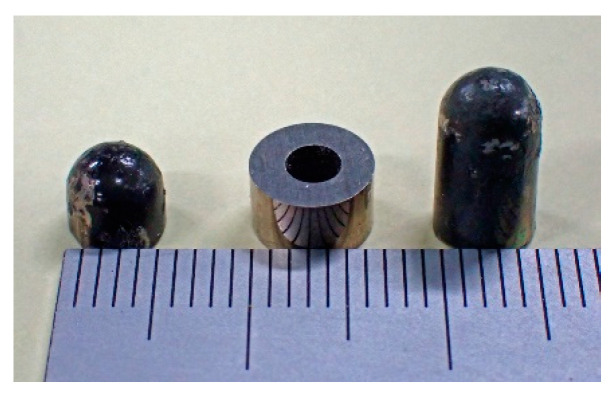
Three different specimens used for the third experiment.

**Figure 25 micromachines-13-00127-f025:**
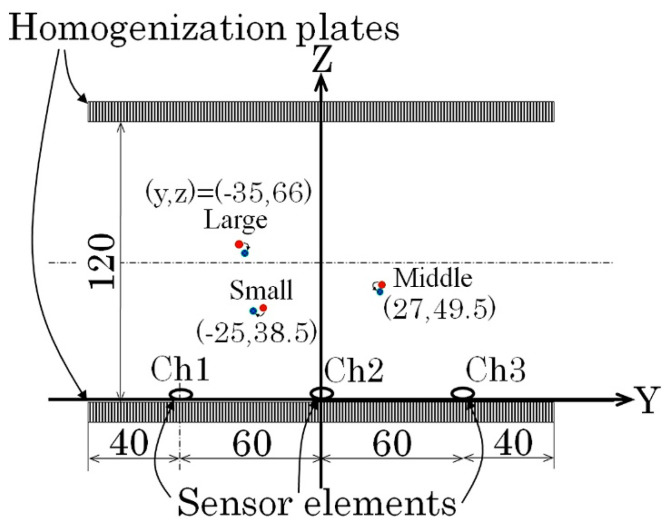
Estimated results for the third experiment (Red: actual position, Blue: estimation).

**Figure 26 micromachines-13-00127-f026:**
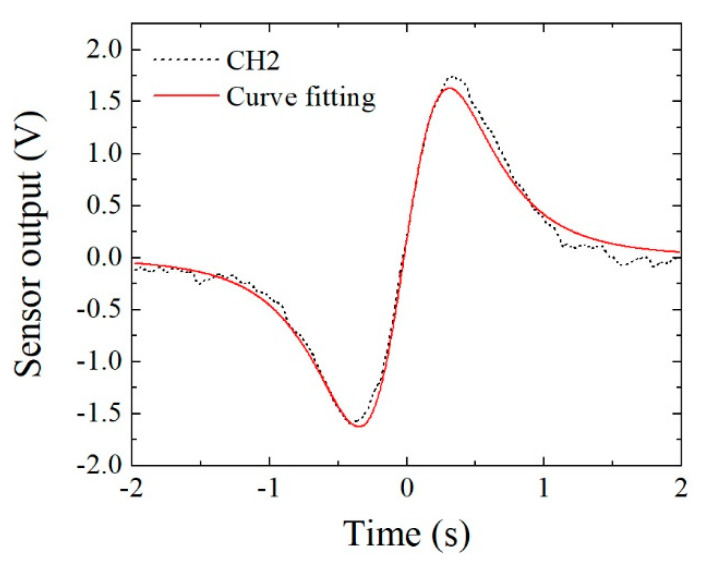
Application of curve fitting using the derived Equation.

**Figure 27 micromachines-13-00127-f027:**
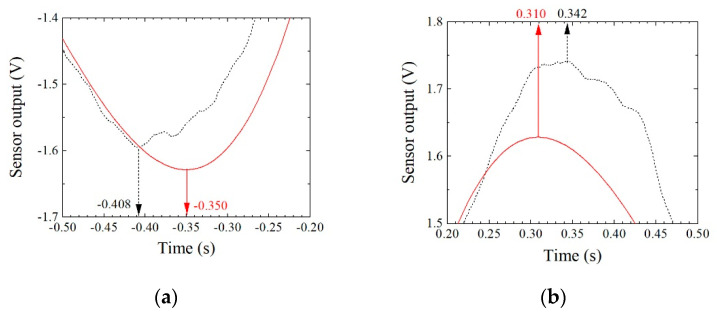
Position difference between raw data and curve-fitting data. (**a**) Minimum point. (**b**) Maximum point.

**Table 1 micromachines-13-00127-t001:** Results of position estimation (second experiment).

Estimated Position (Y, Z)		Actual Position		∆r (Error) (mm)
Y	Z	Y	Z	
41.1	54.9	50	54	8.9
27.1	52.7	30	54	3.2
19.0	51.5	10	54	9.4
−17.7	51.1	−10	54	8.3
−32.8	52.0	−30	54	3.4
−41.3	50.2	−50	54	9.4

**Table 2 micromachines-13-00127-t002:** Results of position and magnetization estimation (Third experiment).

Estimated Position (Y, Z)		Actual Position		∆r (Error) (mm)	Estimated Magnetization: m (Arb.Unit)	Sample No.
Y	Z	Y	Z			
25.5	46.4	27	49.5	3.4	1.2	2
−28.4	37.2	−25	38.5	3.6	1.0	1
−32.5	62.6	−35	66	4.3	4.7	3

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
