# Peer review of "Estimation of Position and Size of a Contaminant in Aluminum Casting Using a Thin-Film Magnetic Sensor"

_micromachines, 2022, doi:10.3390/mi13010127_

Round 1

Reviewer 1 Report

This is a good paper. It describes a system to verify the position of magnetic contaminants in cast pieces. 

The English must be improved very much. It is unacceptable in its present form. Please get a native speaker to read the manuscript and do the articles, prepositions, etc. correctly.

Although there was information about the samples used, I did not see information about the sensors, their dimensions and fabrication. Please furnish this information in the manuscript. How would the results change if the dimensions of the sensor were to change?

How would the results change if the magnetic chips were actually embedded in, for example, an aluminum casting? Would there be eddy currents which would shield the signals? Please discuss.

Reviewer 2 Report

The research is sound and supported by experimental data, but I have some objections to the presentation and some details.

First, I strongly suggest additional editing by a native English speaker (even non-expert). The text is comprehensible overall, but it contains many non-English sentences, in some parts to the point of hindering the presented idea.

Some of the figures (namely Fig. 1, Fig. 9, Fig. 10) are taken without editing from ref. 12. It is OK but should be cited in figure caption – e.g. “Figure 1. Schematic explanation of the concept of measurement system (from [12]).”

Line 134: The extreme position is obtained by the equation dBx/dx=0 using equation (2). – which of the extremes is detected? The minimum, maximum, or whichever comes first? Or midpoint?

Line 162: Fig. 8 . . . magnetic dipole is fed on y=100. – this is probably incorrect. According to text preceding (line 148), the y coordinate is always 50, but z is increased to 100.

Line 231: Figure 13 and Figure 14 - please provide scale for conversion from volts to magnetic units. Also provide information about speed of movement (or comment on scale in terms of X-distance in addition to time units). Otherwise it makes no sense to compare distance between extremes like in Fig.6.

Line 252: Figure 15 shows a dependence of the maximum value of sensor output voltage on the height - how is variation in magnetic properties (mu) and consequently various moment m of chipping compensated? Various materials will provide various strength of response in volts? Or is the height estimation independent from m value? Please comment.

Line 309 - comment: Data in Table 1 may look pessimistic, but when presented graphically in Fig.20 it seems pretty good.

Line 337: central hall – should be central hole (?)

Line 396: Figure 23 shows an example of the curve fitting – ok, but does it provide better position estimation? Can you compare results?

Generally speaking, it may be helpful to provide more detailed (perhaps primitive) derivation of the mathematical formulae for position estimation from input data.

Author Response

Please see attchment.

Round 2

Reviewer 1 Report

The English must be improved before publication,

The authors claim that a certain accuracy was obtained. They should state a value for the accuracy, Be quantitative!!
